# Manufacturing of Corrosion-Resistant Surface Layers by Coating Non-Alloy Steels with a Polymer-Powder Slurry and Sintering

**DOI:** 10.3390/ma16155210

**Published:** 2023-07-25

**Authors:** Grzegorz Matula, Błażej Tomiczek

**Affiliations:** Scientific and Didactic Laboratory of Nanotechnology and Material Technologies, Faculty of Mechanical Engineering, Silesian University of Technology, Konarskiego 18a St., 44-100 Gliwice, Poland; blazej.tomiczek@polsl.pl

**Keywords:** powder metallurgy, pressureless forming, stainless steel, protective coatings, composite

## Abstract

This paper describes the combination of surface engineering and powder metallurgy to create a coating with improved corrosion resistance and wear properties. A new method has been developed to manufacture corrosion-resistant surface layers on steel substrate with additional carbide reinforcement by employing a polymer-powder slurry forming and sintering. The proposed technology is an innovative alternative to anti-corrosion coatings applied by galvanic, welding or thermal spraying techniques. Two different stainless-steel powders were used in the research. Austenitic 316 L and 430 L ferritic steel powders were selected for comparison. In addition, to improve resistance to abrasive wear, coatings containing an additional mixture of tetra carbides (WC, TaC, TiC, NbC) were applied. The study investigates the effects of using multicomponent polymeric binders, sintering temperature, and atmosphere in the sintering process, as well as the presence of reinforcing precipitation, microstructure and selected surface layer properties. Various techniques such as SEM, EDS, hardness and tensile tests and corrosion resistance analysis are employed to evaluate the characteristics of the developed materials. It has been proven that residual carbon content and nitrogen atmosphere cause the release of hard precipitations and thus affect the higher mechanical properties of the obtained coatings. The tensile test shows that both steels have higher strength after sintering in a nitrogen-rich atmosphere. Nitrogen contributes over 50% more to the tensile strength than an argon-containing atmosphere.

## 1. Introduction

The development of modern engineering materials is dependent on and closely related to the technology of forming and sintering powders [1,2,3]. The high requirements set by consumers regarding high properties and low costs make it necessary to look for new technological solutions. Steel is still the best material solution for corrosion-resistant elements and relatively high mechanical loads. Unfortunately, alloying additives that determine high corrosion resistance are expensive [4]. Technologies using anti-corrosion protection, based mainly on zinc and paint coatings, are widely used to coat structural steels. Unfortunately, the influence of zinc galvanization on the hardness of covered steels is significant. For example, many types of steel suffer from a considerable decrease in hardness, particularly high-strength steel [5]. In the analyzed zinc-coated steels, the reduction in hardness ranged from about 28% to as much as 55%.

To obtain a material that is resistant to corrosion and, at the same time, has high mechanical properties, stainless steels with a high chromium and nickel content should be used. Unfortunately, a dynamic increase in nickel prices has been caused by its increasing use. This element’s average annual price growth rate is 7.29% [6]. The demand for nickel also results from its unique chemical properties that make it useful for various applications, like catalysts in methanation [7] or solid oxide fuel cells [8]. Due to the low-temperature coefficient of resistance, nickel-chromium alloys are used in devices operating at high temperatures [9]. It should therefore be expected that the costs of austenitic steels will increase even more. The price of high-nickel steels is four to six times higher than unalloyed steels and twice as high as high-alloy tool steels [4]. Therefore, searching for new materials and technological solutions is important to reduce the share of costly elements such as nickel in steel.

Duplex steels with a ferritic-austenitic structure are undoubtedly an interesting solution. Due to their optimal variety of mechanical properties and high corrosion resistance, duplex steels have more applications. Obtaining high properties is possible from the balance in dual-phase composition and the steel production method, the parameters of individual processes and the insertion of alloying additives. Only the selection and strict control of each listed aspect make it possible to obtain duplex steel that meets the application requirements. Powder metallurgy is widely used in duplex steel production methods [10,11,12,13,14]. Austenitic-ferritic steel powder can be obtained using conventional base powder mixing, compaction and sintering. A powder with a precise chemical composition can be prepared by atomization [15], or base powders with different chemical compositions can be mixed to obtain the correct ratio. In [10], a mixture of austenitic and martensitic corrosion-resistant steel powders was used, and in [11], a combination of austenitic and ferritic powders was used. Various forming techniques are also used. The metal injection molding technique produced small orthodontic components with complex shapes from duplex steel [12]. Often, during the sintering of duplex steel, the chemical composition is equalized during diffusion at high temperatures, and the share of the ferritic phase concerning the austenitic phase increases, as proved in [16]. Duplex steel can also be produced by additive manufacturing. An example of this is a multilayer steel structure made of austenitic and martensitic stainless-steel wires using wire and arc additive manufacturing equipment based on plasma arc welding [17]. An interesting solution may be using powder metallurgy methods for the manufacturing of corrosion-resistant surface layers by coating non-alloy steels. Previous studies [18] have shown that it is possible to produce a layered material with high mechanical strength using polymer-powder slurry.

The main goal of the undertaken research is the development of layered materials resulting from the combination of surface engineering and powder metallurgy. Materials with a layered structure consisting of a stainless surface layer on steel intended for thermal improvement, with high mechanical properties, were developed using polymer-powder slurry and sintering. In addition, hard carbide particles were introduced to the surface layer’s structure to increase the mechanical properties, particularly hardness and resistance to abrasive wear, while maintaining strong corrosion resistance. Particularly noteworthy is the innovative approach to stainless steel, especially 316 L austenitic steel, where it is essential to maintain a low carbon concentration. To prevent intergranular corrosion, it is important to block the precipitation of chromium carbides and the drop in electrochemical potential at the grain boundaries. Because carbon lowers the solidus temperature and initiates the sintering process, which ensures a surface layer’s diffusion connection with the non-alloy steel substrate, a local increase in carbon concentration is required. The proposed technology is an innovative alternative to anti-corrosion coatings applied by galvanic, welding or thermal spraying techniques [19,20]. It is worth emphasizing that the developed method of forming layers resistant to corrosion and wear on steel is innovative and is a unique invention of the authors. So far, there have not been any reports in the literature on using similar solutions.

Surface layers based on ferritic 430 L or austenitic 316 L steel developed as part of this research can be used to cover the screws of extruders and injection molding machines for processing plastics. The increasing use of recyclates with a higher viscosity than pure polymers, which may additionally be contaminated with solid particles, causes an increase in the wear of the surfaces of cylinders and screws. Their regeneration by surfacing with alloys with a high proportion of Co and Ni is associated with high costs. Using the presented technology may reduce these costs and maintain comparable tool properties. This technology is expected to be used primarily in producing new components exposed to wear and corrosion. However, using these layers to regenerate or repair worn steel surfaces may also be technologically and economically justified.

## 2. Materials and Methods

To produce corrosion-resistant surface layers on carbon steels, the powders of austenitic steel 316 L and ferritic steel 430 L marked according to ASTM and manufactured by Sandvik Osprey Ltd. were used. The powders employed had spherical particles typically created by atomizing inert gas [21]. The morphology and particle size distribution are shown in Figure 1. The particle size distribution analysis of the selected powders was performed using the laser particle size analyzer Analysette 22 MicroTec Plus, Fritsch Gmbh, and the results are gathered in Table 1.

The scanning microscope SUPRA 35 by Zeiss (Oberkochen, Germany) was used to determine the powders’ morphology and study the sinters’ structure. ASTM 4140 steel was used as the substrate. Steels used for the surface layer, especially 316 L steel with an austenitic structure, have low hardness, so to increase it, and in particular to increase resistance to abrasive wear, the steel was reinforced by a mixture of carbides known by the name “Tetra Carbides” and produced by Treibacher Industrie AG, containing 47% WC, 14% TiC, 33% TaC and 6% NbC in volume. The Tetra Carbides powder is referred to as TC. The morphology of these carbides and the particle size distribution are also shown in Figure 1. Their volume fraction concerning steel was 5%. The applied powders of stainless steels, atomized by gas, are generally used for the production of feedstock for powder injection molding. A total of 90% of the 316 L and 430 L steel particles are smaller than approximately 19 and 16 µm. Moreover, the particles are spherical, which improves surface wettability with polymers. A total of 90% of the carbide TC particles used to reinforce the stainless steel were smaller than 9 µm. The size distribution of carbide particles is bimodal, which is caused by the strong aggregation of fine carbide particles, and this can be observed in the structure of the sintered samples. To determine the mechanical properties of the surface layers in the form of steel and carbide steels produced on carbon steels, it was necessary to prepare samples of these materials for which the powder injection molding technology was used. The powders of the used stainless steels are suitable for this technology not only due to their spherical shape and size below 20 µm, but also due to their particle size distribution, which is evidenced by the particle size distribution slope parameter S_w_. This parameter is the slope of the log-normal cumulative distribution and can be calculated using Formula (1) [22]. The particle distribution is narrower the higher the value of S_w_. A broad particle size distribution (S_w_ of 2–4) indicates easy-to-mold, low-viscosity material, but a narrow particle size distribution (S_w_ of 4–7) of powder often results in high feedstock viscosity. This ensures high surface quality and sinter edges and, in particular, low surface roughness. Considering the powder parameters (Table 1) and the values of the S_w_ coefficient calculated on this basis, it can be concluded that all the powders used can be used in the powder injection molding technology because the particle size distributions are relatively wide.
(1)$$S_{w}\;=\frac{2.56}{log\frac{D_{90}}{D_{10}}}$$

As part of the preliminary research, various powder-forming techniques were used. As a result of these analyses, the non-pressure forming method was selected as the best due to the properties of the finished element. The technology of forming polymer-powder slips on solid steel surfaces allows for a local increase in the share of carbon initiating the powder sintering process, as well as the surface layer and substrate, which guarantees a good connection of the layer with the diffusion substrate. Figure 2 shows a diagram of this process.

Due to the direction of degradation of the polymer binder from the surface into the layer, the largest share of residual carbon will be found in the area directly above the surface of the substrate. A high proportion of carbon lowers the sintering temperature [23], which reduces the properties of the steel but guarantees good adhesion of the coating to the substrate. An essential issue in this method is the uniform thickness of the layer applied from the polymer-powder slurry. Therefore, it is necessary to introduce automation and control of the coating application process. For example, in the case of components with a circular cross-section, it is possible to dispense the slurry with the simultaneous rotation of the coated bar, which guarantees even distribution of the polymer-powder slurry, as shown in Figure 3.

To perform comparative tests of the mechanical properties of steels and carbide steels, polymer-powder slurries based on a binder containing PP and PW were prepared. A Zamak-Mercator MP-30 mixer was used to homogenize the polymer-powder mixtures. The rotational speed of the mixer screws was 20 rpm, the homogenization temperature was 170 °C, and the time was 30 min. The carbides were pre-mixed with the binder by adding stearic acid as a surfactant, which increases the wettability of the carbide powder surface [24]. Using homogenized slurries, samples for testing were produced using Zamak Mercator equipment. A mini-piston injection molding machine with a cylinder capacity of 15 cm^3^ from the same company was used for injection molding. The actual injection pressure is much higher, but unfortunately, the device cannot measure it. It is only possible to adjust the pressure of the air supplied to the actuator. The injection conditions depended on the shape of the sample. For the beam intended for bending, the conditions were as follows: cylinder temperature 170 °C, die temperature 40 °C, injection time 5 s, pressure 5 bar. In the case of samples with more complex shapes, such as dog bones intended for tensile testing, the viscosity of the slurry should be lower; hence, the temperature of the cylinder and die were 180 °C and 50 °C, respectively, and the other parameters remained the same. A series of samples, such as tensile paddles and beams for the three-point bending test, were thus prepared.

Regardless of the powder-forming method, the produced samples are characterized by a smooth surface. Figure 4a shows a sample in the form of a dog bone injection molded from 316 L steel. The samples are characterized by high quality and a lack of defects in the form of distortions and external and internal bubbles. Figure 4b shows samples of steel 4140, which were covered with a polymer-powder slip and sintered. Samples were made with holes in the centre to test the ability to cover the surface of the inner holes. The drawing shows a clean steel substrate, slip-coated steel, and the finished sinter.

The injection molded samples were then subjected to binder degradation and sintering. The degradation was carried out in two stages. Initially, solvent degradation in heptane was used for max. 24 h, at a temperature of 25 °C. The first step of the degradation allowed paraffin removal. Then, the samples were placed in a tube furnace, in which thermal degradation of the binder and direct sintering was performed at temperatures between 1150 and 1350 °C, with steps of 50 °C. Solvent degradation generally facilitates thermal degradation, the cycle of which has been selected experimentally. Both thermal degradation and sintering were performed in a Czylok tube furnace in an atmosphere of a flowing gas mixture comprised of N_2_-10% H_2_ and Ar-10% H_2_. The maximum heating rate did not exceed 5 °C/min, and during heating, the thermal degradation temperature was much lower and did not exceed 1 °C/min. Due to the high viscosity of the slurries used for injection and significant technological problems with their low-pressure application on the surface of unalloyed steel, a mixture was prepared in which only paraffin was used as a polymer binder. The proportion of steel to carbide powders was comparable. However, in the case of steel powder and steel-carbide powder, the volume fraction of the paraffin binder was raised by 10% and 15%, respectively, to ensure the low viscosity of the slurry. The use of carbides requires a higher proportion of binders due to their small size, irregular shape and the resulting greater specific surface area that needs to be wetted. Table 2 presents the composition of powders and binder slurries intended for injection molding solid samples and forming surface layers on stainless steel. The coated samples were only subjected to thermal degradation at 200 °C for 1 h and then heated directly to the sintering temperature. The sintering time of injection molded and low-pressure samples was 30 min. Injection molded sinters were subjected to shrinkage and density analysis using the hydrostatic method. The microhardness measurement was carried out in the Vickers Future-Tech FM-700 hardness tester with a load of 100 g. The tensile strength and three-point bending tests were performed using appropriate attachments in the Zwick/Roell Z020 testing machine.

Observations of the structure of the produced materials were made in a scanning electron microscope (SEM) ZEISS SUPRA 35, using the detection of secondary electrons and backscattered electrons at an accelerating voltage of 20 kV and a maximum magnification of 50,000×. The corrosion resistance test was conducted on a precise Atlas-Sollich 0531 EU potentiostat according to the PN ISO 17475:2010 standard [25]. In addition to the corrosion tests of sinters, reference samples in commercial steel 316 L and 4140 were also tested. The following parameters were tested: open circuit potential EOCP, corrosion resistance (Ecorr) or breakdown potential (Eb), polarization resistance (Rp), and corrosion current density (icorr). The tribological tests were carried out using equipment for the “ball-on-disc” test, which was performed on a Tribometer CSM. The wear tracks were measured using a Sutronic 25 profilometer from Taylor Hobson and observed on an SEM microscope. A replaceable pin in the form of a small ball with a diameter of 6 mm made from Al_2_O_3_, loaded with 30 N force, was slid on the flat surface of the sample tested. It must be emphasized here that the ball surface wear was negligibly low.

## 3. Results

The solvent degradation of injection molded materials allowed the removal of 98% of the paraffin and facilitated thermal degradation at a later stage. The lack of solvent degradation often causes the formation of gas bubbles on the surface of the sinters. Removal of one of the polymer components, paraffin, allows for the partial opening of the pores in the entire volume of injection molded fittings. In the case of low-pressure molded surface layers, it is necessary to use only thermal degradation of the binder. The test results of injection molded materials show that the shrinkage value after sintering increases with the increase in sintering temperature. In addition, the shrinkage depends on the sintering atmosphere used; in particular, it is more significant for samples sintered in the N_2_-10% H_2_ atmosphere and increases in the case of sinters with additional carbides (Table 3). After sintering at 1250 °C in an Ar-10% H_2_ atmosphere, the shrinkage of 316 L and 430 L steels was 7.5 and 10.3%, respectively, and 9.34 and 11.26% in the N_2_-10% H_2_ atmosphere. Thus, it can be seen that the shrinkage depends on the atmosphere and the type of material. The density of these steels sintered in the Ar-10% H_2_ atmosphere is comparable and amounts to 87.5 and 86.8% for steel 316 L and 430 L, respectively. Using an atmosphere of N_2_-10% H_2_ causes an increase in the density of steel by only about 1.5% in both cases. The results of hardness tests confirm that adding carbides significantly increased the hardness of the tested sinters, but the change in atmosphere did not increase hardness in the case of 316 L steel. In particular, the hardness of 316 L steel after sintering at 1250 °C under Ar-10%H_2_ atmosphere was 198 HV_0.1_, and was comparable when sintered in N_2_-10%H_2_ atmosphere amounting to 196 HV_0.1_. The hardness of this sintered steel was increased to 327 HV_0.1_ by adding carbides at the same temperature and N_2_-10%H_2_ atmosphere. In the case of 430 L steel, the change in the sintering atmosphere had a much greater effect on hardness. After sintering in an argon-rich atmosphere, the hardness was 172 HV_0.1_ and 412 HV_0.1_ when using a nitrogen-rich atmosphere. Adding carbides to this steel increased the hardness to 302 and 546 HV_0.1_ after sintering in the atmosphere of Ar-10%H_2_ and N_2_-10%H_2_, respectively. Therefore, in the case of 430 L steel, the sintering atmosphere is quite essential. Tensile testing has shown that a nitrogen-rich atmosphere increases the strength of 316 L and 430 L steels. In the case of 316 L steel, the change in atmosphere from Ar-10% H_2_ to N_2_-10% H_2_ during sintering at 1250 °C caused an increase in tensile strength from 410 to 643 MPa and a decrease in elongation from 32 to 6.6% (Figure 5). An atmosphere rich in nitrogen undoubtedly strengthens the structure of this steel. It should be noted that the maximum tensile strength of this steel is 652 MPa and can be achieved by adding carbides and sintering in a nitrogen-rich atmosphere. A similar trend can be observed in the case of 430 L steel. Detailed test results are presented in Table 3.

The scanning microscope tests showed that the increase in the tensile strength of the steel is due to the release of carbonitrides after sintering in the N_2_-10%H_2_ atmosphere. In the case of both steels, they are rich in Cr and Fe, which was revealed by scanning microscopy and EDS analysis, as shown in Figure 6 and Figure 7. Determining whether Mo and Ni are also part of these precipitates in 316 L steel is difficult because the phases are less than 1 µm. Similar phases precipitate in 430 L steel sintered in a nitrogen-rich atmosphere. The structure of both steel grades resembles a pearlitic structure due to the presence of fine precipitates of nitrides. Tribology studies (Figure 8) have shown that 316 L stainless steel sintered in an N_2_-rich atmosphere achieves significantly higher abrasion resistance than steel sintered in an Ar-rich atmosphere. This is undoubtedly the effect of the precipitated nitrides. The width of the trace of abrasion of the sample sintered in a mixture of Ar-10%H_2_ gases is 1739μm, and the depth is 62.4 μm. For the material sintered in the atmosphere of N_2_-10%H_2_, these values are 564 and 11.8 μm, respectively. The depth of the abrasion trace of steel sintered in a nitrogen atmosphere is more than five times lower, which is a surprising result for the authors. Table 4 shows the calculated volume of material removed in the ball-on-disc test. Both materials sintered in a nitrogen-rich atmosphere had lower wear compared to steels sintered in an argon-rich atmosphere.

Figure 9 shows carbide steels 316 L/TC and 430 L/TC produced by injection molding powders and sintered at 1250 °C. Comparing the structure of both materials, it can be seen that similar phases are separated in both materials. Grey carbides rich mainly in Cr, Fe and W are marked in both carbon steels as No. 2. Light carbides are rich mainly in W but also in Fe and Cr labelled as No. 3. There are also fine dark precipitates rich in nitrogen, carbon, and oxygen and titanium in these materials. Due to the small particle size, the Cr and Fe content may partly come from the matrix. The chromium concentration in the 430 L/TC carbide matrix is lower than that in the 430 L steel powder, and its mass fraction is 13.9 and 16.5%, respectively. Similarly, in 316 L/TC carbide, chromium concentration in the matrix decreases from 17.3 to 15.2%. This results from the precipitation of carbides rich in this element and the depletion of the matrix, which is a typical effect in stainless steels with a high concentration of carbon. Corrosion tests were also performed on injection molded samples because their surface is flatter and regular. The results of corrosion tests (Table 5) have shown that sintering in a nitrogen-rich atmosphere and adding carbides increases sinters’ corrosion resistance. This effect is quite surprising, but it is most likely due to the lower porosity of these materials. In general, residual carbon, a nitrogen-rich atmosphere and other carbides should have the opposite effect, i.e., corrosion resistance should be lower. The residual carbon and nitrogen from the sintering atmosphere give off phases rich in chromium, which are responsible for corrosion resistance. However, it should be noted that both residual carbon and nitrogen cause sinter densification.

The mixture of carbides used has a regular crystalline structure, which is stable at high temperatures. Only WC carbide crystallizes in a hexagonal lattice. It dissolves at high temperatures in the matrix of 316 L or 430 L stainless steel and, unfortunately, forms new carbides often rich in Cr, which was revealed in the tests performed in SEM and using EDS. Adding carbides to 316 L steel sintered in an argon-rich atmosphere makes parameters such as E_ocp_, E_kor_, and E_b_ better than 4140 steel, but the corrosion current density is more than twice as high. The best anti-corrosion properties were found in the sinters produced for 316 L steel at the same sintering atmosphere without adding carbons.

Analyzing the individual parameters, the free potential of the E_ocp_ material and the corrosion resistance of the E_corr_ material is better for 430 L steel with the addition of carbides and a nitrogen-rich atmosphere. At the same time, E_b_ is worse except for the 430 L/TC/N_2_-10%H_2_ material. The tests of injection molded samples and their results confirmed the reasonableness of producing surface layers of these materials on a non-alloy steel substrate. The drawings presented below result from a microscopic examination of selected examples of surface layers. Figure 10 and Figure 11 are particularly noteworthy, proving that a good connection with the diffusive substrate should characterize these layers.

Figure 11 shows an enlargement of the area of the surface layer-substrate boundary with a clear diffusion zone in which there are no carbides but an apparent increase in the concentration of chromium and nickel. It is a layer with a thickness of approx. 15 mm, which separates the substrate from the layer of 316 L steel. The structure of the layer in this figure is rich in carbides, which confirms the thesis that the concentration of residual carbon in this area should be high, which initiates the sintering process of the surface layer with the substrate. Observing the structure of the surface layer on the surface of the sample, it can be seen that it is more porous. Therefore, paraffin degradation in this area is more accessible, and the proportion of residual carbon is lower or absent. Also, the share of carbides in this area is smaller, as evidenced by the Cr concentration distribution presented in the linear distribution of elements. Unfortunately, the porous structure may reduce corrosion resistance. However, it is still higher than the substrate, as confirmed by Figure 12 and Figure 13 which show the surface layer observed under a light microscope before and after etching with Nital. The base structure in 4140 steel is less resistant to Nital.

Tests of surface layers reinforced with carbides showed that their structure is dense, with few pores that can be observed just below the surface of the layer. Figure 14 shows the 430 L/TC layer sintered at 1250 °C in an N_2_-10% H_2_ atmosphere. Comparing PIM and PLF, it can be seen that a heterogeneous structure with numerous carbide agglomerates characterizes the produced surface layers enriched with carbides. This results from the manual preparation of the slurry in a mortar. The injection molded material is more homogeneous because although local agglomerates can also be seen, they are not as numerous as in the low-pressure molding method. Undoubtedly, the mixture prepared in the crusher is more homogeneous. Agglomerated carbides are also seen in the bimodal particle size distribution (Figure 1). To continue this research, attention should be paid to homogenizing the structure.

## 4. Discussion

A newly created technique has been developed that uses polymer-powder slurry forming and sintering to create corrosion-resistant surface layers on steel substrates with additional carbide reinforcement. The materials fabricated this way are characterized by high quality and a lack of defects like distortions and bubbles inside and outside. Injection molded materials’ solvent degradation enabled almost all paraffin removal and later promoted heat degradation. A lack of solvent degradation often causes the formation of gas bubbles on the surface of the sinters. This results from accumulating gaseous thermal degradation products formed during pyrolysis [26,27]. The pores in the total volume of injection molded samples can be partially opened by removing one of the polymer components, i.e., paraffin. However, the low-thickness surface layers require only the binder’s thermal decomposition [18].

According to geometry measurement results of samples produced through injection molding, the shrinkage value grows with increasing sintering temperature. Even though no quantitative porosity test was performed, the shrinkage results can be considered indirect information about the compaction of the material during sintering. Unfortunately, the decrease in porosity is not entirely dependent on the increase in the tested density. As a result of sintering in an atmosphere rich in nitrogen, this gas diffuses into the sample, and fine precipitations are in the form of nitrides. The precipitation process affects the compaction kinetics and thus affects the density of the material. Of course, the sintered sample density also increases due to the addition of carbide phases. Generally, 316 L austenitic steel is characterized by lower shrinkage than 430 L ferritic steel and correspondingly higher porosity, which is consistent with the results of other authors [28]. According to a comparison of the shrinkage and density of sintered materials in various atmospheres, the gas used has little influence on the degree of densification of the structure of the produced materials. In both instances, using an atmosphere of N_2_-10% H_2_ only results in a 1.5% increase in steel density.

The results of hardness tests have shown that materials sintered at 1250 °C are characterized by higher hardness than commercial steels [29]. Hardness investigations show that adding carbides greatly increased the hardness of the studied samples, but 316 L steel’s hardness was not affected by changes in the gas atmosphere. The difference in sintering atmosphere had a substantially more significant impact on hardness in the case of 430 L steel. Of course, adding carbides increases the hardness to a maximum value of 546 HV0.1 for the 430 L/TC coating sintered in a nitrogen-rich atmosphere.

However, the observations are different regarding the influence of the atmosphere on strength properties. The tensile strength of 316 L steels sintered in the Ar-10% H2 atmosphere was below the strength of commercial steels and had a similar elongation value. According to tensile tests, a nitrogen-rich atmosphere improves the strength of 316 L and 430 L steels. In contrast, the use of N_2_-10%H_2_ atmosphere caused an increase in strength properties to a value exceeding commercial 316 L steel, i.e., 643 MPa and, unfortunately, a decreasing elongation by nearly five times. This effect corresponds with other authors’ investigations [30]. Without a doubt, the nitrogen-rich atmosphere strengthens the structure of stainless steel, which has already been well documented in the literature [31,32,33]. Unfortunately, all sinters produced are characterized by lower corrosion resistance than commercial steel 316 L, and in the case of 316 L/TC material sintered in an Ar-10% H2 atmosphere, resistance was even worse than the support material. Analyzing the influence of the atmosphere and the share of carbides, a general conclusion can be drawn that the corrosion resistance of these materials depends mainly on porosity.

Furthermore, according to tribology experiments, 316 L stainless steel sintered in an N2-rich atmosphere exhibits significantly better abrasion resistance than steel sintered in an Ar-rich atmosphere. Beyond any uncertainties, this results from the precipitated nitrides, mainly Cr_2_N [30]. The wear track of the 316 L sample sintered in an argon-rich atmosphere showed a combination of adhesion, abrasion, and plastic deformation. Small metallic fragments appeared shortly after the test began, unlike the samples sintered in nitrogen gas, where friction produced barely any tiny particles. Visible changes in the friction force suggested that the steel and ball were sticking together and causing adhesive wear. On the sample sintered in Ar+10%H_2_ gas, severe adhesive and abrasive wear was observed, whereas the specimens sintered in N_2_+10%H_2_ revealed only very mild abrasive wear and some plastic deformation as the layer was pressed into the substrate at higher loads. Differences in the behavior of nitrogen-enriched stainless-steel samples are consistent with reports in [34].

Corrosion studies have demonstrated that sintering in an atmosphere rich in nitrogen and adding carbides improved the corrosion resistance of sintered samples. The fact that tests were conducted on the surface of sintered samples that had not been ground or given any other kind of treatment suggests that this impact, however unexpected, is caused by the lower porosity of these materials. The corrosion resistance should generally be reduced in the presence of residual carbon, a nitrogen-rich atmosphere, and other carbides [35]. Chromium-rich phases are produced by the excess carbon and nitrogen from the sintering atmosphere, which prevents corrosion. However, it should be emphasized that sinter densification is brought on by both residual carbon and nitrogen. Nitrogen contributes indirectly to the compaction of the sinter by forming nitrides or carbonitrides, which causes the released non-carbide-forming carbon to initiate the sintering process [36]. Undoubtedly, the lower porosity of sinters increases corrosion resistance, which is confirmed by the results of other authors [37]. The legitimacy of using nitrogen in austenitic stainless steels is indisputable. It can replace expensive nickel. Nitrogen stabilizes the austenitic structure and improves mechanical and anti-corrosion properties, but unfortunately, it reduces plasticity [38,39,40,41,42,43,44,45]. Due to the low solubility of nitrogen in Fe in conventional steels, it is introduced during their melting under high pressure. Some alloy additions also improve the solubility of nitrogen in steel [39]. Sintering these steels in a nitrogen-rich atmosphere is a better solution. Unfortunately, Cr nitrides are released during sintering, which has been confirmed by test results and data in the literature [46]. Similar phases can be seen in Co-Cr-Mo alloys sintered in a nitrogen-rich atmosphere [47]. The latest research results confirm the separation of these phases and, additionally, a decrease in the sinterability of steel, which increases porosity, which in turn decreases mechanical properties and corrosion resistance [32]. Interpretation of some test results is quite difficult. For example, a nitrogen-rich sintering atmosphere for 316 L steel increases tensile strength and wear resistance while maintaining the hardness at the level of steel sintered in an Ar-rich atmosphere. This requires further research of these materials, mainly using higher sintering temperatures, which will increase the density and mechanical properties of the sinters and, most likely, increase corrosion resistance.

The pressureless forming technology enabled the application of polymer-powder slurries containing stainless steels and their mixtures with carbides on the base of unalloyed steels. The structure of the surface layers is similar to the previously produced sinters. A diffusion layer can be observed in the boundary zone with the substrate, which is rich in alloy additions typical of stainless steels, such as Cr and Ni, which diffuse into the substrate made of unalloyed steel. This area is characterized by low porosity. Many carbides can also be observed in this zone, although they were not added to the polymer-powder slurry. This is the effect of an increase in the concentration of carbon, which initiates the sintering process but simultaneously causes the precipitation of Cr-rich carbides. The surface of the produced top layer is characterized by much greater porosity. It can be lowered by adding a mixture of tetra carbides to the slurry, thus increasing the hardness. Irrespective of the type of low-pressure polymer-powder slurry, the sintering surface layers are characterized by an excellent diffusive connection with the substrate and do not show discontinuities in the form of cracks and decohesion. Automating the process of applying the polymer-powder slurry onto elements with a round cross-section or an extended surface is possible. The materials produced this way are also characterized by a continuous surface layer but with a different thickness and structure. Therefore, further research should be conducted to automate the process of forming anti-corrosion or anti-wear surface layers.

## 5. Conclusions

The aim of the research was to develop layered materials using polymer-powder slurry molding and sintering, which resulted in the creation of steel with a durable, corrosion-resistant surface layer. Several conclusions can be made considering all of the observed results:Sinters such as steels 316 L, 430 L and carbide steels with a matrix of stainless steels, produced by injection molding of powders, are characterized by relatively low porosity, depending mainly on the sintering temperature.The results of hardness tests have shown that materials sintered at 1250 °C are characterized by higher hardness than commercial steels, regardless of the atmosphere used and the addition of carbides.The higher mechanical properties of the obtained stainless-steel coating were influenced by the increase in carbon concentration resulting from residual carbon and nitrogen in the sintering atmosphere, which caused the release of carbonitrides, regardless of the steel grade.Tensile testing shows that 316 L and 430 L steels have higher strength after sintering in nitrogen-rich atmospheres. The atmosphere changes from Ar-10% H_2_ to N_2_-10% H_2_ during sintering improved tensile strength by more than 50% and decreased elongation by almost five times. Nitride precipitations undoubtedly strengthen the stainless steel’s structure.

## Figures and Tables

**Figure 1 materials-16-05210-f001:**
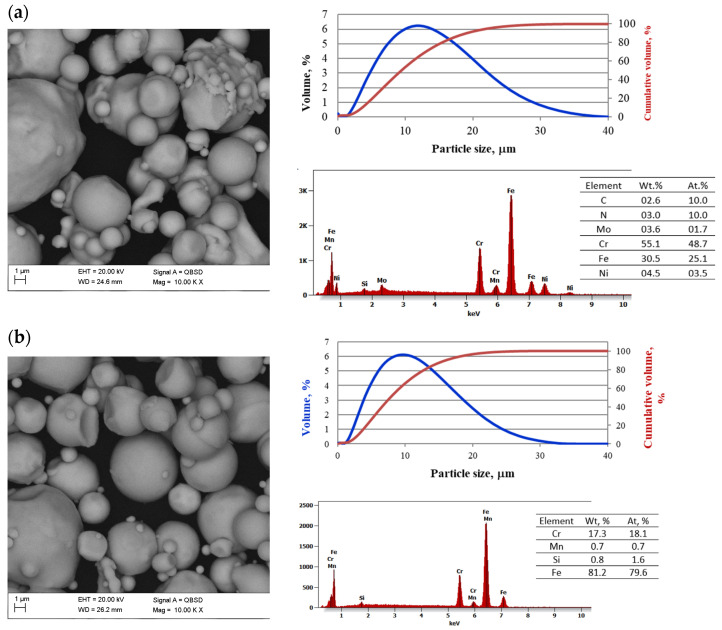
SEM morphology, particle size distribution and EDS chemical composition of (**a**) 316 L steel powder, (**b**) 430 steel powder (**c**) tetra carbides powder. Particle size distribution curve is blue and cumulative curve is red.

**Figure 2 materials-16-05210-f002:**
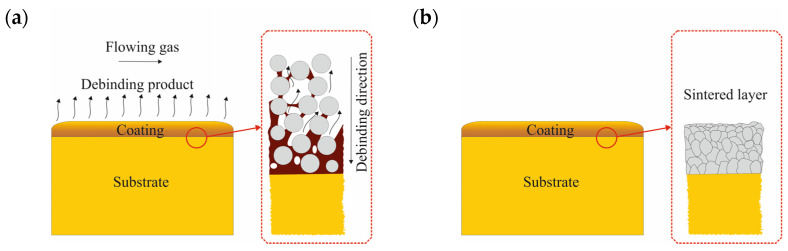
Scheme of (**a**) thermal debinding process of polymer-powder coatings applied on a solid steel substrate and (**b**) manufactured coating with stainless steel sintered layer.

**Figure 3 materials-16-05210-f003:**
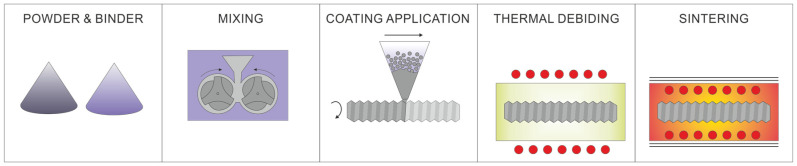
Scheme of the automation process of applying polymer-powder coatings.

**Figure 4 materials-16-05210-f004:**
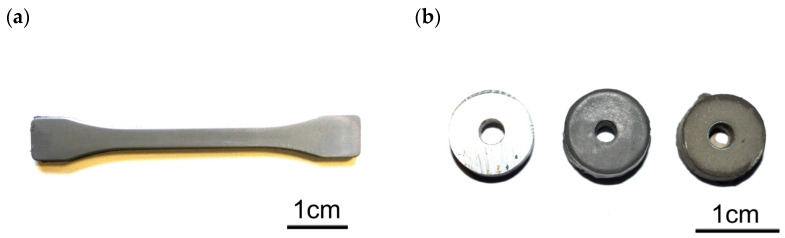
View of (**a**) a 316 L steel sample after powder injection molding and (**b**) pre-coating, post-coating and post-sintering samples with holes.

**Figure 5 materials-16-05210-f005:**
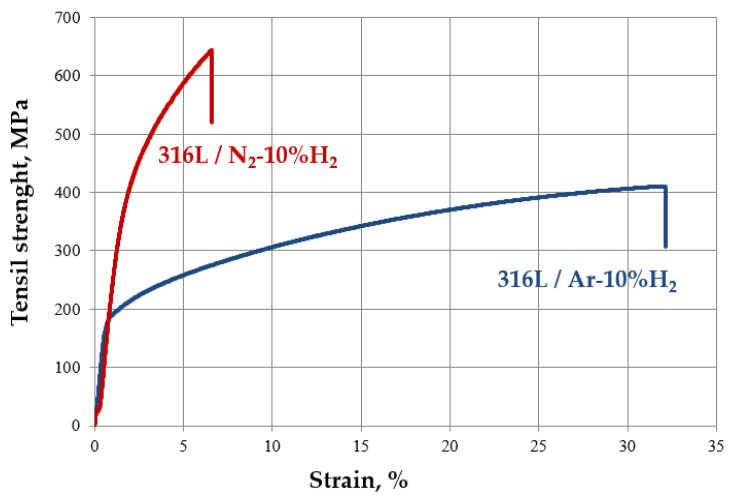
Measured stress–strain curves for 316 L steel sintered under Ar-10%H_2_ and N_2_-10%H_2._

**Figure 6 materials-16-05210-f006:**
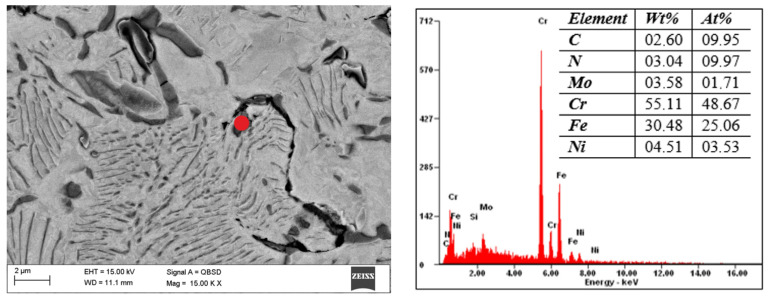
Structure of 316 L steel sintered under N_2_-10%H_2_ at a temperature of 1250 *°*C and chemical composition of (marked with red dot) precipitation of investigated materials observed in SEM.

**Figure 7 materials-16-05210-f007:**
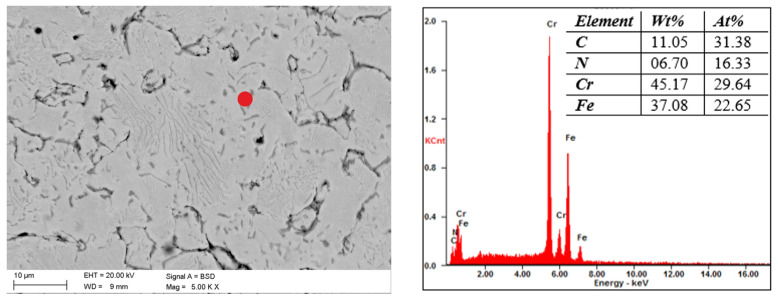
Structure of 430 L steel sintered under N_2_-10%H_2_ at a temperature of 1250 *°*C and chemical composition of (marked with red dot) precipitation of investigated materials observed in SEM.

**Figure 8 materials-16-05210-f008:**
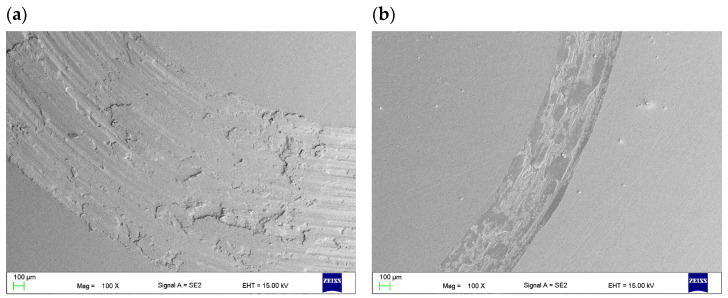
Wear track of the 316 L steel sintered under (**a**) Ar-10%H_2_, (**b**) N_2_-10%H_2_ and its depth profiles (**c**,**d**), respectively.

**Figure 9 materials-16-05210-f009:**
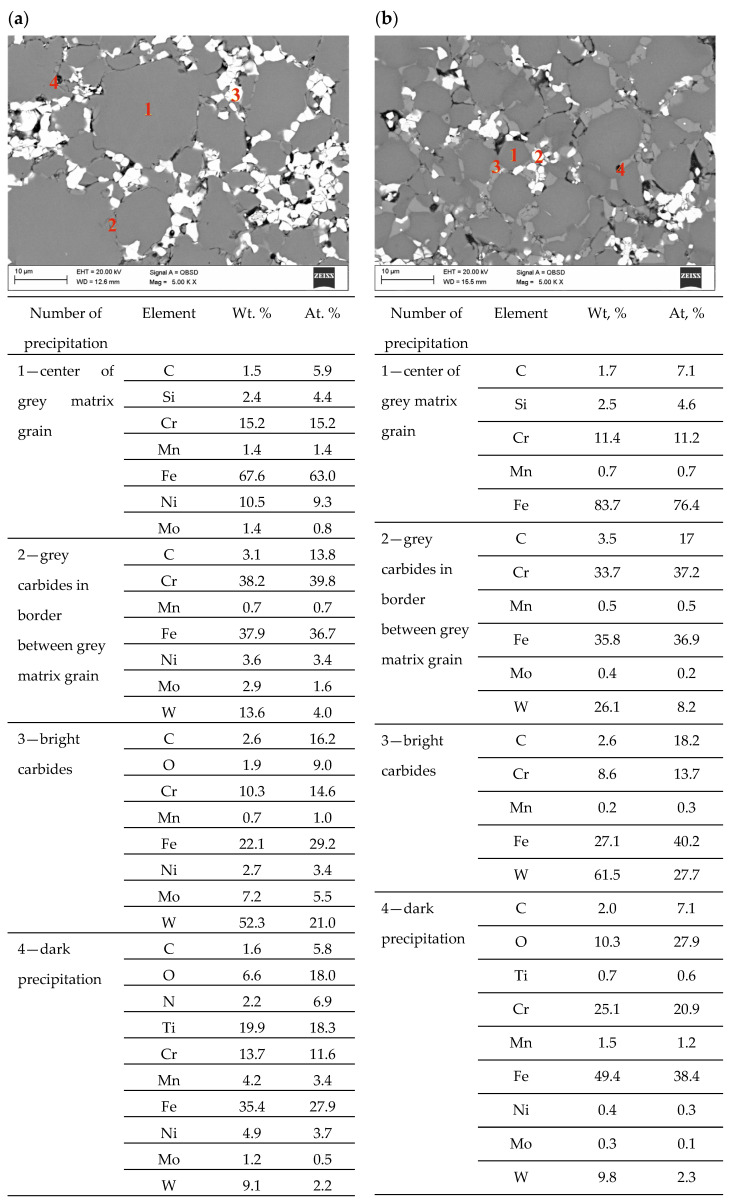
Structure of carbide steel sintered at 1250 °C and chemical composition of precipitation of investigated materials observed in SEM, (**a**) 316 L/TC/N_2_-10%H_2_, (**b**) 430 L/TC/N_2_-10%H_2_.

**Figure 10 materials-16-05210-f010:**
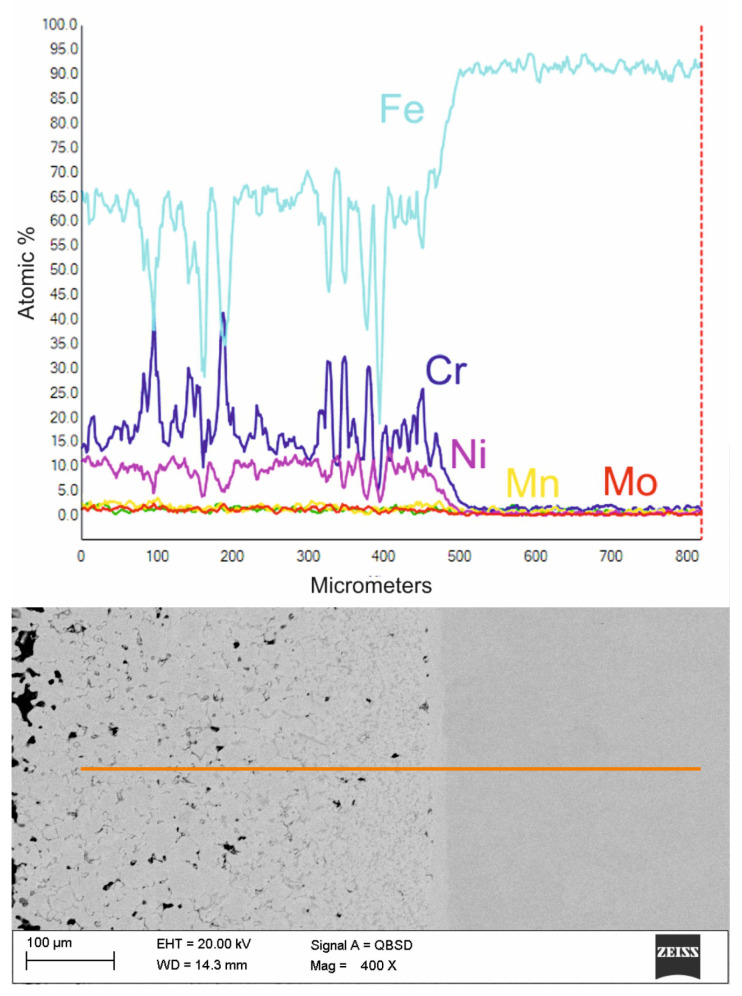
The structure of the surface layer of 316 L steel, manufactured by slurry pressureless forming on the 4140 substrate and sintered at 1250 °C in an atmosphere of N_2_-10%H_2_ and the distribution of elements in the area marked with a line.

**Figure 11 materials-16-05210-f011:**
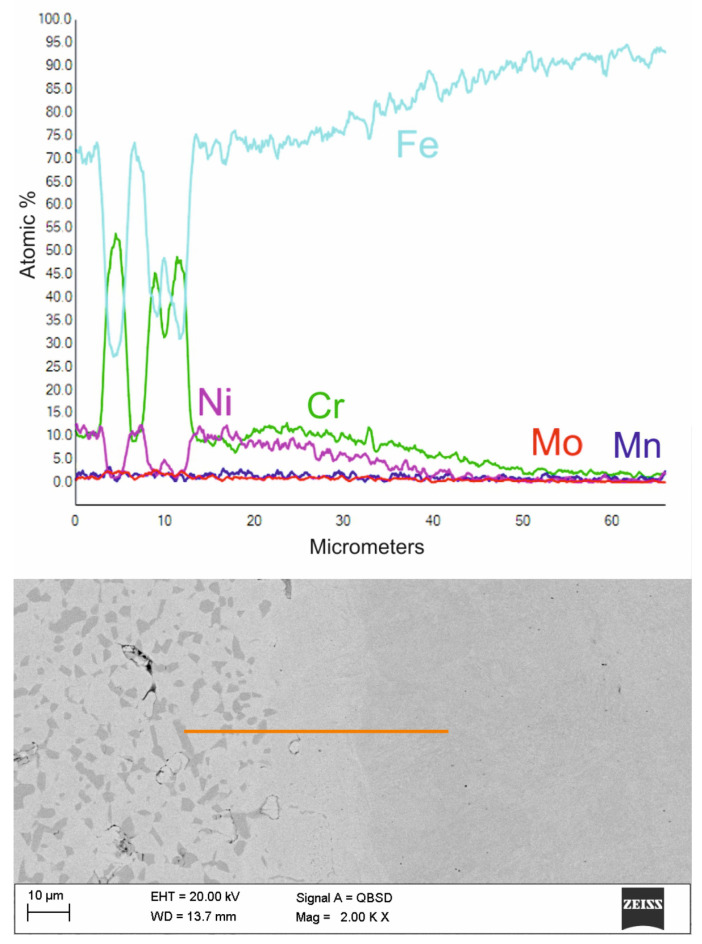
The structure of the surface layer made of 316 L steel, manufactured by slurry pressureless forming on the 4140 substrate and sintered at 1250 °C in an atmosphere of N_2_-10%H_2_ and the distribution of elements in the area marked with a line—higher magnification of boundary area.

**Figure 12 materials-16-05210-f012:**
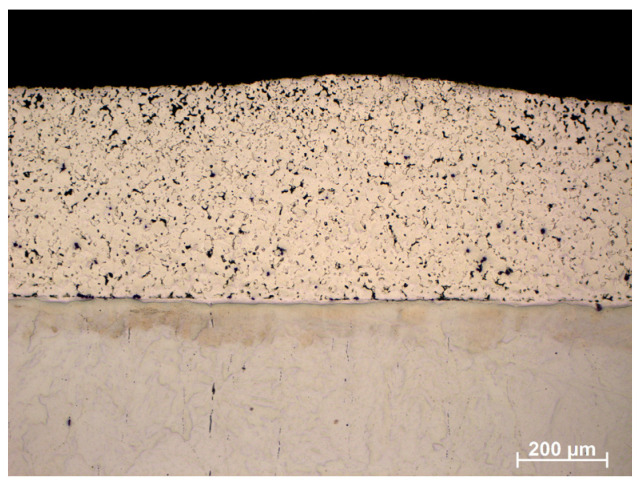
Structure of the surface layer made of 316 L steel, manufactured by slurry pressureless forming on the 4140 substrate and sintered at 1250 °C in N_2_-10%H_2_ atmosphere—light microscope.

**Figure 13 materials-16-05210-f013:**
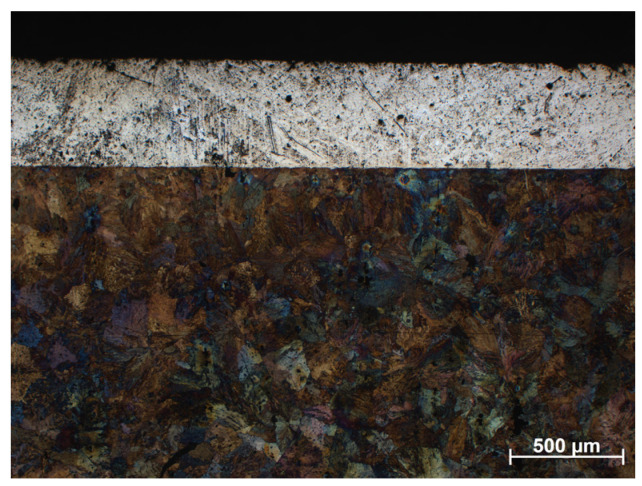
The structure of the material shown in Figure 12 etched with Nital.

**Figure 14 materials-16-05210-f014:**
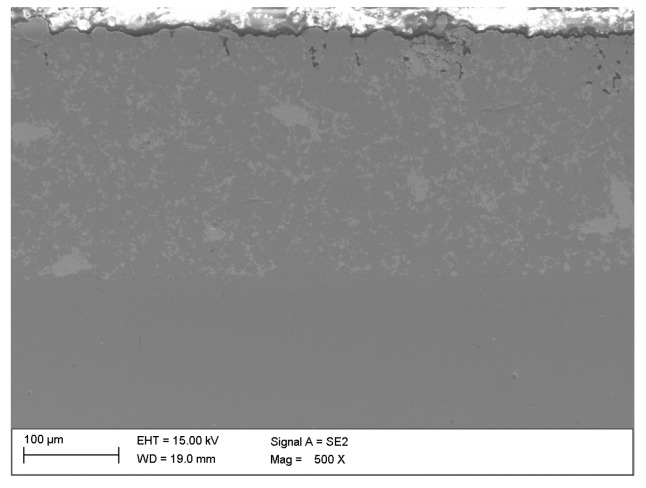
Layer structure of 430 L/TC steel, manufactured by slurry pressureless forming on unalloyed steel and sintered at 1250 °C in Ar-10%H2 atmosphere.

**Table 1 materials-16-05210-t001:** Characteristics of base powders.

Powder	316 L	430 L	Tetra Carbides
Density, g·cm^−3^	7.94	7.70	11.82
D10, µm	3.78	3.19	0.80
D50, µm	9.88	8.16	2.70
D90, µm	19.99	16.83	9.48
Sw	3.53	3.54	2.74

**Table 2 materials-16-05210-t002:** Compositions of injected and pressureless formed samples.

Molding Methods	Designation	Powder Volume Fraction, %			Binder Volume Fraction, %			Density of Slurry, g·cm^−3^
		316 L	430 L	Tetra Carbides	PW	SA	PP	
Powder Injection Molding PIM	PIM316	60	-	-	20	-	20	5.12
	PIM430	-	60	-	20	-	20	4.98
	PIM316TC	54	-	6	19.8	0.4	19.8	5.35
	PIM430TC	-	54	6	19.8	0.4	19.8	5.22
Pressureless forming of powder PLF	PLF316	50	-	-	50	-	-	4.42
	PLF430	50	-	-	50	-	-	4.3
	PLF316TC	40.5	-	4.5	55	-	-	4.27
	PLF430TC	-	40.5	4.5	55	-	-	4.17

**Table 3 materials-16-05210-t003:** Properties of obtained materials sintered at 1250 °C under different atmospheres.

Material	316 L		316 L/TC		430 L		430 L/TC	
Atmosphere	Ar-10%H_2_	N_2_-10%H_2_	Ar-10%H_2_	N_2_-10%H_2_	Ar-10%H_2_	N_2_-10%H_2_	Ar-10%H_2_	N_2_-10%H_2_
Density	6.92	7.03	6.98	7.027	6.693	6.79	7.68	7.825
Shrinkage	7.54	9.34	9.24	10.05	10.32	11.26	11.12	11.28
Hardness, HV_0.1_	198	196	217	327	172	412	302	546
Tensile strength, MPa	410	643	559	652	432	668	658	679

**Table 4 materials-16-05210-t004:** Influence of sintering atmosphere of stainless steel on its wear after ball-on-disc test.

Material	316 L Ar-10%H_2_	316 L N_2_-10%H_2_	430 L Ar-10%H_2_	430 L N_2_-10%H_2_
Volume of wear material, µm^2^	1.86	0.106	1.48	0.52

**Table 5 materials-16-05210-t005:** Corrosion test results for investigated materials.

	E_ocp_, mV	E_kor_, mV	E_b_, mV	J_kor_, µA/cm^2^	R_pol_, kΩ × cm^2^
316 L comparative material	−132	−160	344	0.09	171
316 L/Ar-10%H_2_	−286	−294	25	0.54	46
316 L/N_2_-10%H_2_	−303	−310	−196	18.1	1
316 L/TC/Ar-10%H_2_	−506	−511	−420	56	0.5
316 L/TC/N_2_-10%H_2_	−349	−370	−118	5	4.4
430 L/Ar-10%H_2_	−469	−434	−157	6.1	3.8
430 L/N_2_-10%H_2_	−300	−425	−256	6.6	2.9
430 L/TC/Ar-10%H_2_	−423	−429	−265	5.2	3.7
430 L/TC/N_2_-10%H_2_	−363	−369	−149	3.3	5.8
4140 substrate	−602	−590	−517	25	0.8

## Data Availability

Not applicable.

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
