# Peer review of "Manufacturing of Corrosion-Resistant Surface Layers by Coating Non-Alloy Steels with a Polymer-Powder Slurry and Sintering"

_materials, 2023, doi:10.3390/ma16155210_

Round 1

Reviewer 1 Report

This manuscript used a polymer powder slurry to coat unalloyed steel and sinter it to prepare a corrosion-resistant surface layer. The reviewer considers that the technical content of this manuscript meets the quality requirements of the journal, but that the presentation must be improved. The readability of the manuscript was low, full of lengthy sentences and only one paragraph per section, making it difficult for the reader to follow. The manuscript must be significantly improved in its narrative before it can be accepted. In addition, some comments are as follows:

-         In the introduction section, the description of the necessity and innovation of this study needs to be enhanced. Also, perhaps a separate paragraph would make this section clearer.

-         Lack of necessary dimensioning in Fig. 4.

-         L149-191, L217-252 were full of lengthy narratives, please refine them and add to the readability of this manuscript in a separate paragraph.

-         The quality of figures 7 and 8 requires improvement, note the red wavy line below Wt.

The readability of the manuscript was low, full of lengthy sentences and only one paragraph per section, making it difficult for the reader to follow. 

Author Response

Dear Reviewer

Thank you for your review. We have corrected our paper according to your suggestions.

In the introduction section, the description of the necessity and innovation of this study needs to be enhanced. Also, perhaps a separate paragraph would make this section clearer.

Based on your suggestion, we've improved the introduction. We have put a separate paragraph with information about the research's main goal and novelty.

- Lack of necessary dimensioning in Fig. 4.

According to your request, we have add dimensioning to the figure 4.

- L149-191, L217-252 were full of lengthy narratives, please refine them and add to the readability of this manuscript in a separate paragraph.

Based on your suggestion, we've improved the section 2 and 3.

- The quality of figures 7 and 8 requires improvement, note the red wavy line below Wt.

According to your request, we have improved the quality of the figures.

- The readability of the manuscript was low, full of lengthy sentences and only one paragraph per section, making it difficult for the reader to follow.

The current version of the manuscript has been spell checked. We have divided sections to paragraphs.

We are grateful for your contribution to our research development.

Reviewer 2 Report

Please read the attachment. Thank you.

Minor changes in grammar and structure are needed.

Author Response

Dear Reviewer 

Thank you for your review. We have corrected our paper according to your suggestions.

- Abstract: Please add a short statement of research gaps and the trends of the fields in the concluding sentences.

Based on your suggestion, we've improved the abstract. We actually rewrote it completely.

- Keywords: Please provide between 5 and 10 keywords that should not repeat the words/phrases that appeared in the manuscript title.

According to your suggestion, we have changed the keywords.

- Introduction: please add a paragraph to introduce the outline of the manuscript.

Based on your suggestion, we've improved the introduction

- All equations should be mentioned or explained in the main text.

According to your request, we have add explaination of equations Sw

- Please revise all units. for example, g/cm3 should be g.cm-3 etc.

According to your request, we have changed units

- Table 2: please remove the bolded phrases in the title.

Done

- Table 3: please remove the bolded phrases in the title.

Done

- Figure 6: its quality is too poor. Please increase its resolution.

According to your request, we have improved the quality of the figure 6

- Figure 9: please change "Fig." to "Figure." Please add the titles and units for the vertical and horizontal axes.

Done

- Literature review: Manufacturing of corrosion-resistant surface layers and the related ones. The following work could be helpful.

According to your request, we have add a few new references.

In addition, we provide answers to your constructive questions:

  1. How does the choice of polymer type and its concentration in the slurry affect the resulting carbon concentration after thermal degradation? What are the specific degradation conditions investigated, and how do they influence the carbon content?

The decomposition rate of polymers depends primarily on their physical and chemical properties. Amorphous polymers are more susceptible to degradation than crystalline polymers. Polymer binders with a simple structure are degraded faster than branched ones. The polymers with higher molecular mass are characterise by slower decomposition. For example, the molecular mass of paraffin is much lower than that of polypropylene. To minimise the residual carbon, paraffin was employed, which has a short, straight chain and is readily thermally degraded. In addition, the paraffin's low melting point and, in particular, the low viscosity facilitates mixing with the powder and formation of the top layer.

  1. Could you provide more details on introducing hard particles into the surface layer during the homogenization of the polymer-powder slurry? What are the characteristics of these particles, and how do they enhance the wear resistance of the developed materials?

Particles of steel powders and carbides mixed with liquid paraffin were homogenized in a ceramic mortar for 15 minutes. WC hexagonal lattice carbides dissolve at high temperatures, reduce porosity and form complex carbides M6C and others rich in Fe and Cr, which reduces corrosion resistance. For this reason, a mixture of regular lattice carbides was used, which are more stable at high temperatures and do not dissolve in the matrix and block grain growth.

  1. In discussing the diffusion connection between the applied polymer powder coating and the non-alloy steel substrate during sintering, could you elaborate on the specific mechanisms that facilitate this connection? How does the carbon concentration at the interface play a role in ensuring a strong diffusion connection and preventing delamination?

The degradation of the binder, which is directed perpendicular to the substrate's surface, causes the local increase in carbon content at the surface layer-substrate boundary. The binder degrades from the surface into the surface layer. The zone located on the border with the substrate degrades the slowest. Hence the local increase of residual carbon initiating sintering occurs. A higher local concentration of dissolved carbon in the solid solution of iron lowers the melting point and thus the diffusion sintering temperature. It can be seen that the porosity of the applied top layer decreases towards the coated steel substrate. The distribution of carbon concentration is shown in the graph resulting from the EDS analysis.

I hope we dispelled your doubts and you are satisfied with our answer. We are grateful for your contribution to our research development.

Reviewer 3 Report

The research aims to develop layered materials by combining surface engineering and powder metallurgy, resulting in steel with a durable, corrosion-resistant surface layer. This alternative method avoids welding or thermal spraying techniques. The study focuses on using multiple polymeric materials as binders for the slurry formed on a steel substrate, with subsequent removal through thermal degradation. The concentration of residual carbon, which increases during degradation, depends on the type of polymer used, its proportion, and degradation conditions. The presence of carbon is desirable as it initiates the sintering process. Particularly, at the interface between the polymer powder coating and the non-alloy steel substrate, a significant increase in carbon content facilitates a strong diffusion connection and prevents delamination. Furthermore, the research explores the introduction of hard particles resistant to abrasive wear into the surface layer. These particles are incorporated during the homogenization of the polymer-powder slurry, with stainless steel powder as the primary component. This addition enhances the wear resistance of the investigated materials. The investigations employed techniques such as SEM (Scanning Electron Microscopy), EDS (Energy-Dispersive X-ray Spectroscopy), microhardness tests, and assessments of mechanical properties and corrosion resistance to analyze the surface layers.

The paper provided contains information related to the development of modern engineering materials, particularly focusing on the use of powder metallurgy and the need to reduce the reliance on costly elements such as nickel in steel. However, there are several areas where the whole paper could be improved:

In the introduciton part:

The introduction lacks clarity in its presentation of the main objectives and focus of the research. It covers a broad range of topics, including the use of nickel in various applications, the demand for electric cars, and the high cost of high-nickel steels. The introduction should provide a clear and concise statement of the research objectives and its relevance.

The introduction includes some information that is not directly relevant to the research topic, such as the use of nickel in batteries and its presence in car batteries. These details distract from the main focus of the research and should be omitted or provided in a more concise manner.

In the discussion part:

The discussion primarily focuses on presenting the research findings without providing sufficient context or background information. It would be beneficial to include a brief introduction or recap of the research objectives and methodology to help readers understand the significance of the findings.

Some statements lack sufficient explanation or clarification, making it difficult for readers to fully grasp the meaning or significance. Providing more detailed explanations and elaborating on complex concepts or terms would improve the clarity of the discussion.

By addressing these points, the introduction and discussion sections could be enhanced to provide a clearer, more comprehensive understanding of the research findings and their implications.

Author Response

Dear Reviewer 

Thank you for your review. We have corrected our paper according to your suggestions.

The introduction lacks clarity in its presentation of the main objectives and focus of the research. It covers a broad range of topics, including the use of nickel in various applications, the demand for electric cars, and the high cost of high-nickel steels. The introduction should provide a clear and concise statement of the research objectives and its relevance. The introduction includes some information that is not directly relevant to the research topic, such as the use of nickel in batteries and its presence in car batteries. These details distract from the main focus of the research and should be omitted or provided in a more concise manner.

Based on your suggestion, we've improved the introduction. We removed unnecessary information not related to the subject and modified this section to make it more readable.

The discussion primarily focuses on presenting the research findings without providing sufficient context or background information. It would be beneficial to include a brief introduction or recap of the research objectives and methodology to help readers understand the significance of the findings. Some statements lack sufficient explanation or clarification, making it difficult for readers to fully grasp the meaning or significance. Providing more detailed explanations and elaborating on complex concepts or terms would improve the clarity of the discussion. By addressing these points, the introduction and discussion sections could be enhanced to provide a clearer, more comprehensive understanding of the research findings and their implications.

Based on your suggestion, we've improved the discussion. We have provided brief introduction and more explanations.

Reviewer 4 Report

Review report: Manufacturing of corrosion-resistant surface layers by coating non-alloy steels with a polymer-powder slurry and sintering. Work is presented well with good publishing quality and can be accepted after the following corrections:  

1.       Abstract: Add some quantitative results related to mechanical testing at end of the abstract section.

2.       Introduction: In place of citing multiple references, explain the individual work of the author and try to make a bridge between current and previous work. Refer to some recently published work: https://doi.org/10.1007/s12540-020-00705-w.

3.       Novelty and application: Add a separate section for novelty and application of work.

4.       Materials and methods: The section is presented well but needs some corrections. Add a detail of the experimental setup instead of a schematic image. Also, add a reference for Table 1. Plate after coating is not presented. The purpose of selecting different metal powder for coating is also not clear. If possible add EDS result of the particle. Fig. 4 should come in the experimental section and also add the standard used for specimen preparation.

5.       Purpose of measuring shrinkage is not clear. Also mention the method used for preparation.

6.       Add the image of the fractured specimen and also combine both the tensile plot.

7.       Add detail wear mechanism. It is difficult to get any technical information from the current discussion: https://doi.org/10.1007/s12633-017-9710-2.

8.       Combine Fig. 12 and Fig. 13. Also provide the information related to elemental distribution.

9.       There are too many images. Remove the unnecessary images.

10.    Discussion section is very poor. Add previous work and mechanism to relate the current work.

11.    I am really afraid with the conclusion section. This is not acceptable. Keep only 4/5 bullet points in this section.

12.    References are ok.

NA

Author Response

Dear Reviewer

Thank you for your review. We have improved our article according to most of your suggestions.

  1. Abstract: Add some quantitative results related to mechanical testing at end of the abstract section.

Based on your suggestion, we've improved the abstract.

  1. Introduction: In place of citing multiple references, explain the individual work of the author and try to make a bridge between current and previous work. Refer to some recently published work:https://doi.org/10.1007/s12540-020-00705-w.

According to your suggestion, we've improved the introduction and add new references.

  1. Novelty and application: Add a separate section for novelty and application of work.

Based on your suggestion, we have added a new section about the primary goal, novelty and potential application.

  1. Materials and methods: The section is presented well but needs some corrections. Add a detail of the experimental setup instead of a schematic image.

Thank you for your comment, but as the authors, a schematic diagram (Fig. 3) of layer fabrication is needed to improve the article's readability for non-specialists. In the materials and method section, we described the manufacturing method in detail and included the equipment used.

 Also, add a reference for Table 1.

The data collected in Table 1 are the results from our particle size distribution analysis which was made using a laser particle size analyser Analysette 22 MicroTec Plus. You can find the reference in the text to the table in the L110 and L138.

Plate after coating is not presented.

The view of the steel component in the form of disc with a hole after coating and sintering is shown in Fig. 4.

The purpose of selecting different metal powder for coating is also not clear.

The project aims to create layers resistant to corrosion and abrasive wear. Hence two different stainless steel powders were used as a base. Austenitic 316L steel powders and 430L ferritic steel powders were selected for comparison. Due to the different chemical compositions, the indirect goal was to determine how a given steel would harden with nitride precipitates by sintering in a nitrogen-rich atmosphere.

If possible add EDS result of the particle.

According to your request, an EDS result of the powders has been added to Figure 1.

Fig. 4 should come in the experimental section and also add the standard used for specimen preparation.

As suggested, we have moved Figure 4 and a description fragment to the materials and method section. For the preparation of samples for microscopy analysis, we used the instructions found in the struers guides entitled: “Metallographic preparation of powder metallurgy parts” and “Metallographic preparation of stainless steel” from www.struers.com

  1. Purpose of measuring shrinkage is not clear. Also mention the method used for preparation.

We have added the information about reason of shrinkage analysis. The results of shrinkage tests indirectly inform us about the compaction of the material that has not been subjected to quantitative porosity tests. Hence its measurement is justified. Unfortunately, the decrease in porosity is not entirely dependent on the increase in the tested density because, as a result of sintering in an atmosphere rich in nitrogen, this gas diffuses into the sinter from the atmosphere and new phases in the form of nitrides are separated in the material, which increases the density of the sinter. The sintered sample density also increases due to the addition of carbide phases.

  1. Add the image of the fractured specimen and also combine both the tensile plot.

We have combined the two tensile plots in one graph. Unfortunately, we cannot add a photo at this stage because the tensile samples have been cut and included for different tests after the test.

  1. Add detail wear mechanism. It is difficult to get any technical information from the current discussion:https://doi.org/10.1007/s12633-017-9710-2.

Based on your suggestion, we have improved the wear description.

  1. Combine Fig. 12 and Fig. 13. Also provide the information related to elemental distribution.

According to your suggestion, we have combined an SEM photo of the microstructure with a diagram of the distribution of chemical elements.

  1. There are too many images. Remove the unnecessary images.

As recommended, we have removed Fig. 15 and its description.

  1. Discussion section is very poor. Add previous work and mechanism to relate the current work.

Based on your suggestion, we've improved the discussion section.

  1. I am really afraid with the conclusion section. This is not acceptable. Keep only 4/5 bullet points in this section.

According to your suggestion, we have reduced the conclusions to a few points.

The current version of the manuscript has been spell checked. 

I hope we dispelled your doubts and you are satisfied with our answer. We are grateful for your contribution to our research development.

Round 2

Reviewer 4 Report

Accepted. 

NA